# Telescoping bimodal latent Dirichlet allocation to identify expression QTLs across tissues

Ariel DH Gewirtz[1] , F William Townes[2], Barbara E Engelhardt[2,3]

**Expression quantitative trait loci (eQTLs), or single-nucleotide polymorphisms that affect average gene expression levels, provide important insights into context-specific gene regulation. Classic eQTL analyses use one-to-one association tests, which test gene–variant pairs individually and ignore correlations induced by gene regulatory networks and linkage disequilibrium. Probabilistic topic models, such as latent Dirichlet allocation, estimate latent topics for a collection of count observations. Prior multimodal frameworks that bridge genotype and expression data assume matched sample numbers between modalities. However, many data sets have a nested structure where one individual has several associated gene expression samples and a single germline genotype vector. Here, we build a telescoping bimodal latent Dirichlet allocation (TBLDA) framework to learn shared topics across gene expression and genotype data that allows multiple RNA sequencing samples to correspond to a single individual's genotype. By using raw count data, our model avoids possible adulteration via normalization procedures. Ancestral structure is captured in a genotype-specific latent space, effectively removing it from shared components. Using GTEx v8 expression data across 10 tissues and genotype data, we show that the estimated topics capture meaningful and robust biological signal in both modalities and identify associations within and across tissue types. We identify 4,645 cis-eQTLs and 995 trans-eQTLs by conducting eQTL mapping between the most informative features in each topic. Our TBLDA model is able to identify associations using raw sequencing count data when the samples in two separate data modalities are matched one-to-many, as is often the case in biological data. Our code is freely available at https://github.com/gewirtz/TBLDA.**

## Introduction

Genomic differences, such as single-nucleotide polymorphisms (SNPs), among individuals are important drivers of gene expression variability. Much previous work has focused on discovering expression quantitative trait loci (eQTLs), which capture associations between the number of copies of a minor allele present at a given genomic locus and the expression level of a single gene (GTEx Consortium, 2017; GTEx Consortium, 2020). However, a one-to-one mapping of genes to SNPs is too simplistic given the reality of biological interactions and the availability of many observations per individual. Pleiotropy, gene regulatory networks with biological redundancy and feedback loops, and linkage disequilibrium (LD) blocks of highly correlated SNPs all contribute to a complex and dynamic biological regulatory system.

From a statistical perspective, performing genome-wide one-to-one association tests yields an astronomical multiple-testing burden for trans-eQTLs, where the agnostic approach examines every interchromosomal gene and SNP combination. Statistical power is further reduced because trans-eQTLs, or eQTLs where the regulatory SNP is on a different chromosome than the gene that it regulates, often have smaller effect sizes than cis-eQTLs or eQTLs where the regulatory SNP is local to the gene that it regulates (Petretto et al, 2006). One method to reduce the effective number of tests is to cluster correlated SNPs and genes and compare the averaged cluster signals versus testing for every possible marginal association.

Probabilistic topic models, such as latent Dirichlet allocation (LDA), are unsupervised machine learning methods that were initially introduced in natural language processing (Blei et al, 2003) and in statistical genetics as models of ancestry (Pritchard et al, 2000). LDA finds latent topics via soft clustering of feature counts over many samples while simultaneously estimating each sample's topic membership proportions. More recently, these types of models have been applied to gene expression data with gene counts as features. The topics estimated by these models represent interpretable underlying biology such as cell type or developmental stage and have been used in QTL mapping as the quantitative traits themselves (Hore et al, 2016; Dey et al, 2017).

We hypothesized that multimodal topic modeling could identify clusters of covarying genes and SNPs. Existing methods have used Dirichlet process mixture models to integrate two data modalities (Savage et al, 2010), but nonparametric Bayesian models tend to be

---

[1]Lewis-Sigler Institute of Integrative Genomics, Princeton University, Princeton, NJ, USA    [2]Department of Computer Science, Princeton University, Princeton, NJ, USA    [3]Gladstone Institutes, San Francisco, CA, USA

Correspondence: barbara.engelhardt@gladstone.ucsf.edu

too computationally intense for larger data sets such as modern genotype arrays, which capture millions of SNPs. Argelaguet et al (2018) designed a factor model framework (MOFA) to jointly model multiple data modalities, allowing various data likelihoods via link functions. However, relevant methods assume that the modalities are measured on the same samples such that there is a single observation from each individual in each modality (Savage et al, 2010; Virtanen et al, 2012; Zhao et al, 2016; Argelaguet et al, 2018; Li & Gaynanova, 2018; Ash et al, 2021). Many earlier methods also require gene expression data to be normalized, potentially adulterating true signals or spuriously adding false ones (Robinson et al, 2010; Love et al, 2014; Hore et al, 2016; Hicks et al, 2018).

In this work, we create a probabilistic model to find shared structure between gene expression and genotype data. Our model uses raw sequencing read counts and is designed for a nested data structure where although samples are paired, modalities may have different numbers of samples from each subject. This is often the case when we have many samples of gene expression from a particular donor—as in the GTEx data with multiple tissue samples per donor and also for single-cell RNA sequencing samples with multiple cells per subject—but a single germline genotype vector. We apply our unsupervised model to GTEx v8 data and use known sample tissue labels and cell type enrichment scores post hoc to interpret the biological context of the estimated components (GTEx Consortium, 2020). To demonstrate the model's ability to find shared variation between data modalities, we conduct eQTL mapping using the most informative features in each topic to find both known and novel—and tissue-specific and general—cis- and trans-eQTLs.

## Results

We applied our telescoping bimodal LDA (TBLDA) model to gene expression data for the 10 tissues with the highest number of samples from the v8 GTEx data release and to the genotypes from all individuals who contributed to at least one of the samples (Table 1). We took advantage of the known GTEx covariates to interpret biological variation captured in the model factors and ensure relevant signal was found.

First, we checked that ancestral structure, using reported ancestry as a proxy, was not associated with the estimated shared factors. As expected, ancestral structure was largely controlled for because it is captured in the genotype-specific portion of the model (median absolute value factor-ancestry point biserial correlation coefficient < 0.01). This contrasts with the genotype-specific factors, each of which had a point biserial correlation coefficient of at least 0.41 (ranging to 0.99) with at least one reported race.

Next, we looked for signal from one of the top sources of known variation in the data set, tissue of origin, by identifying factors active in specific tissues. We found 15,439 tissue-factor associations via the inner product of each factor and a tissue indicator vector, considering inner products greater than 40 to be tissue-associated (Fig S1 and Table 1). Tissue sample size was strongly correlated with the number of tissue-associated factors (Kendall's rank correlation $\tau$ = 0.64, $P$ < 0.01), which suggests that certain tissues may have underpowered downstream analyses. Overall runs, whole blood, and skeletal muscle had the most associated factors (2,599 and 2,649, respectively), whereas the tibial nerve had the fewest (905). The skin (sun-exposed), skin (not sun-exposed), lung, subcutaneous adipose, and thyroid had the weakest associations (median inner products between 60.7 and 69.7); whole blood, tibial nerve, and esophagus mucosa had the strongest (median inner products between 88.6 and 109.6). Accordingly, whole blood and skeletal muscle samples allocate most of their topic membership into tissue-specific factors (Fig 1).

To explore the robustness of these tissue-associated factors, we compiled sets of the top-ranked features that frequently appeared across factors associated with a common tissue (see the Extended Methods section). Taken together, the two skin tissues had the largest group with 6,515 genes, whereas whole blood had the largest number of unique genes considering the other sets (1,983). All tissue-associated robust gene sets were enriched for functionally relevant biological process Gene Ontology (GO) sets (Benjamini Hochberg [BH] FDR < 0.1, Table S1). Furthermore, all relevant robust tissue gene sets (except for tibial artery) contained most of the tissue-specific transcription factors (TFs) present in the overall analysis (whole blood 10/12, thyroid 6/8, esophagus 7/8, skins 12/14, lung 3/4, nerve 3/3, skeletal muscle 12/12, tibial artery 0/2,

**Table 1. Overall summary statistics and available data for each of the 10 tissues included in the analysis.**

| Tissue | Sample size | Num. tissue-associated factors | Cell type enrichment scores |
|---|---|---|---|
| Subcutaneous adipose | 581 | 1,536 | Adipocytes |
| Tibial artery | 584 | 1,504 | |
| Esophagus mucosa | 497 | 1,254 | Keratinocytes, epithelial cells |
| Lung | 515 | 1,095 | Epithelial cells |
| Skeletal muscle | 706 | 2,649 | Myocytes |
| Tibial nerve | 532 | 905 | |
| Skin (not sun-exposed) | 517 | 1,225 | Keratinocytes, epithelial cells |
| Skin (sun-exposed) | 605 | 1,390 | Keratinocytes, epithelial cells |
| Thyroid | 574 | 1,282 | Epithelial cells |
| Whole blood | 670 | 2,599 | Neutrophils |

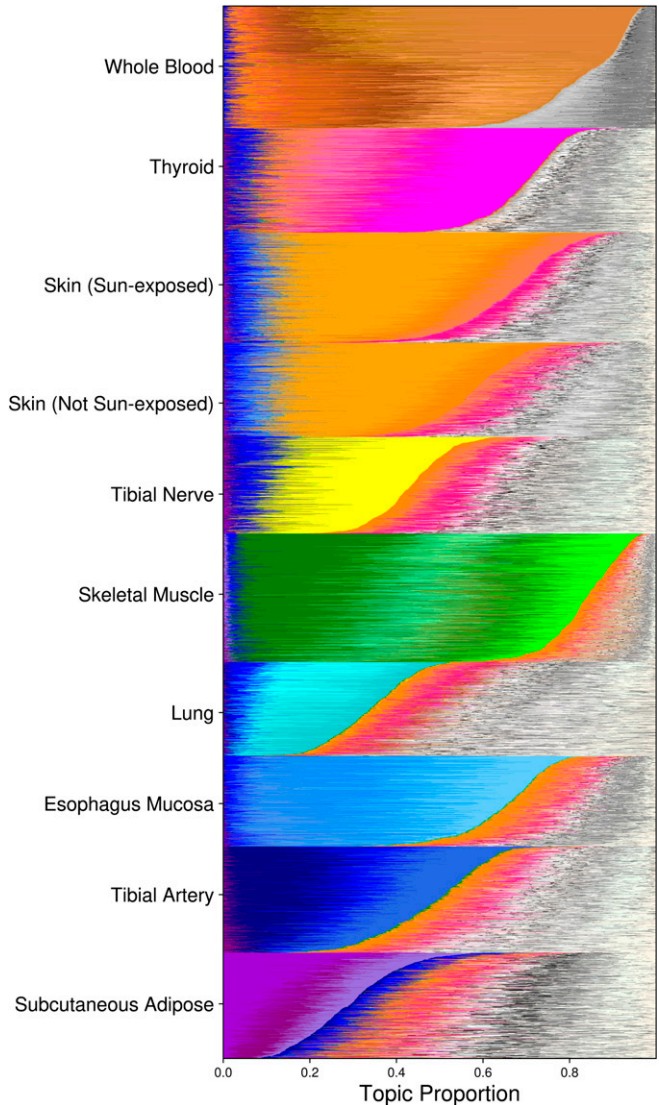

**Figure 1. The estimated TBLDA topics capture tissue-specific signal.**
Each row depicts the expected sample-topic proportion for one sample for the model fit using genes on chromosome 19 and single-nucleotide polymorphisms on chromosome 22; samples are sorted by tissue. Tissue-associated topics are colored in a tissue-wise family color scheme. The remaining topics are drawn using a random gray scale.

subcutaneous adipose 2/2; see the Extended Methods section), whereas only one gene set included another tissue's TF (subcutaneous adipose contained one TF from whole blood) (Sonawane et al, 2017). We note that the tissue-specific TFs were also determined using GTEx data, and our analysis leads to results consistent with these prior results on these same GTEx data. This demonstrates that our model consistently found topics that captured important tissue-specific biological variation including functional pathways and tissue-specific regulatory activity.

Next, we used a compilation of 63 SNP classes describing general annotations shared across tissues from the LDSC (Bulik-Sullivan et al, 2015) data repository to explore functional regulatory enrichments among tissue-associated SNPs. The union of all robust

tissue-associated SNPs was enriched for 20 SNP classes (Fisher's exact test, BH FDR < 0.1) including TFBS ENCODE (BH FDR < 0.016), SuperEnhancer Hnisz (BH FDR ≤ $3.4 \times 10^{-3}$) and active enhancer-associated H3K27ac Hnisz (BH FDR ≤ $9.8 \times 10^{-5}$), and H3K4me1 Trynka (BH FDR ≤ $2.6 \times 10^{-5}$). In particular, several single tissue-associated SNP sets are associated with DGF ENCODE, DHS_Trynka.extend.500, H3K4me1 Trynka, H3K27ac Hnisz, and Enhancer Hoffman (eight, six, five, five, and three tissues, respectively, of the 10 total; BH FDR < 0.1; Fig 2 and Table S2). These SNP set enrichments from our model show that TBLDA identifies functional connections between genotype and gene transcription; these enrichments are intriguing because trans-eQTLs are known to be associated with enhancer activity (GTEx Consortium, 2017).

We then investigated whether the SNP sets are clustered together in particular genomic regions. The union of all tissue-associated SNPs was not enriched in any chromosomal regions using a bin size of 250,000 bp and a sliding window of 100,000 bp, but there were 143 tissue-specific genomic bin enrichments (Fisher's exact test, FDR < 0.1; Fig 2 and Table S3). Notably, 48 regions on chromosome four were enriched for the robust SNP set associated with subcutaneous adipose (BH FDR < 0.05). This highlights the ability of TBLDA to identify jointly functional genomic regions even when the SNP data have been LD-pruned.

Although the GTEx data provide the ground truth of each sample's origin tissue, this is not the case across all data sets. Thus, we next evaluated whether our model could recover robust components across relevant runs in an unsupervised manner. To do this, we ran our model 484 times, once for each pair of chromosomes in the GTEx v8 data, and identified shared components across these runs (see the Extended Methods section). Across all runs, we recovered 197 clusters of robust genotype factors and 1,799 groups of robust gene expression factors. Loadings that were well-correlated with each other across runs tended to cluster by tissue; 81 of the robust genotype clusters and 468 of the robust gene expression clusters included factors that were associated with the same tissue (Fig 3). Only 14 and 75 of the robust genotype and expression clusters, respectively, did not include tissue-associated factors. The presence of these tissue-associated robust genotype components demonstrates that TBLDA identifies interactions between the data modalities versus separate structure within each modality.

Because of the nature of bulk RNA-seq expression data, the GTEx samples average expression over heterogeneous tissue samples containing various cell types. We computed the Kendall correlation between cell type enrichment scores and factor values to determine whether factors capture sample cell type composition (Fig S2). We use estimated enrichment scores for bulk cell deconvolution across five cell types (adipocytes, keratinocytes, epithelial cells, myocytes, and neutrophils) in 8 tissues for a total of 11 tissue and cell type pairs (Table 1) (GTEx Consortium, 2020). Enrichment scores for 8/11 pairs of tissue and cell types were well captured by at least one factor (maximum abs (Kendall $\tau$) > 0.5). This suggests that the TBLDA components often represent cell type–specific processes within tissue samples.

To test whether traditional eQTLs ascertained using univariate tests are captured by TBLDA, we ran a linear model for association between the top 10% most informative SNPs and genes on common

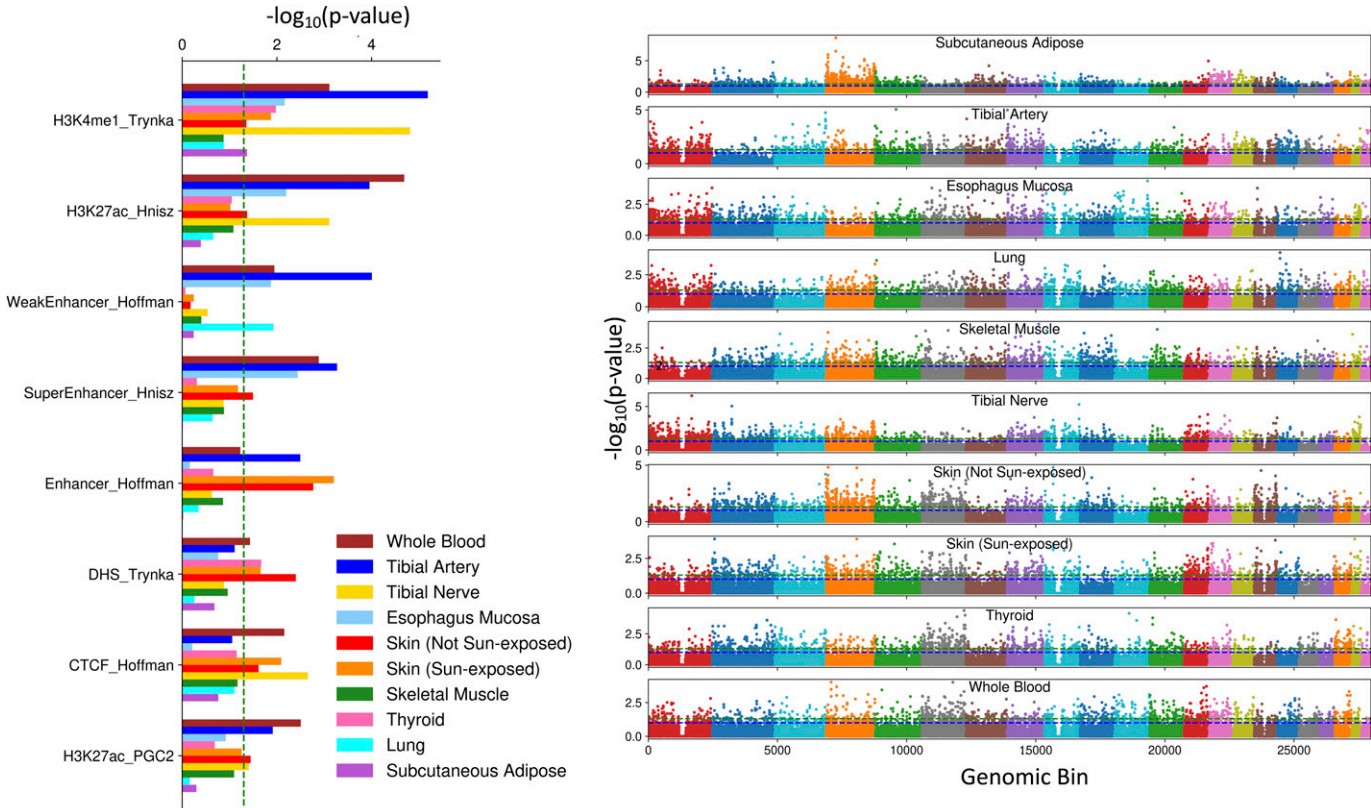

**Figure 2. Robust tissue-associated single-nucleotide polymorphism (SNP) sets are enriched for DNA markers and localized throughout the genome.**
Left: enrichment via Fisher's exact test of eight of the 63 LDSC SNP classes across all robust tissue-associated SNP sets. Right: each tissue's associated SNP set was tested for genomic localization via Fisher's exact test. The blue and green dotted lines are drawn at *P*-value thresholds of 0.1 and 0.05, respectively. The colors mark the division between ordered chromosomes, with chromosome one on the far left.

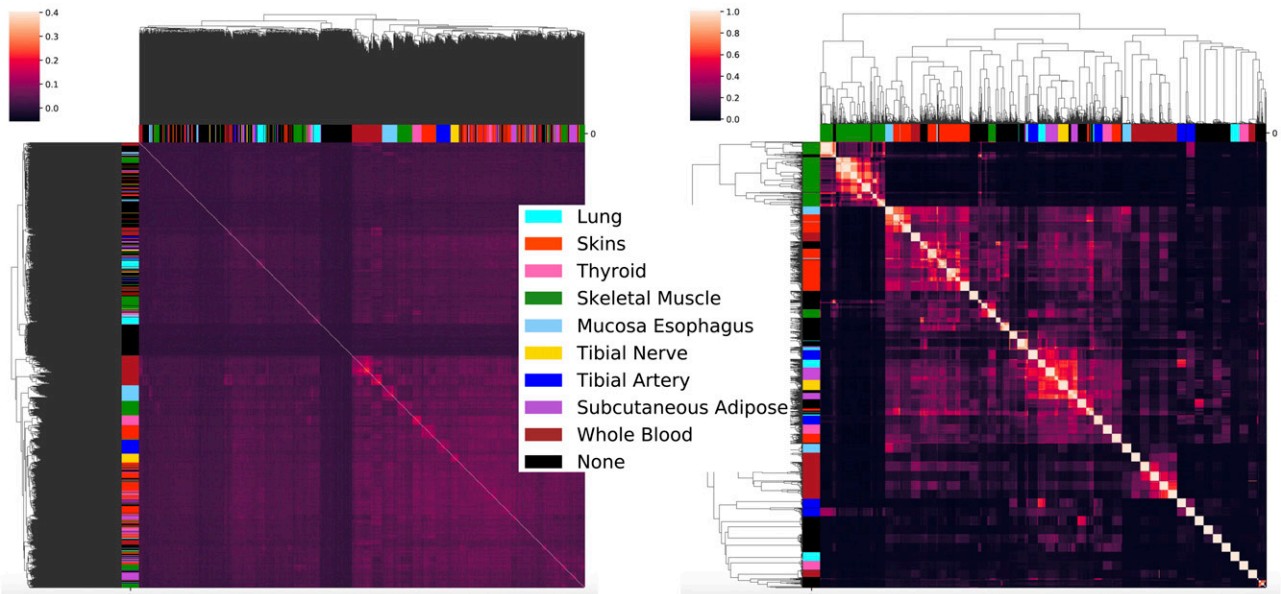

**Figure 3. The TBLDA model estimates robust factors across independent runs.**
Cluster maps of the pairwise Pearson correlations between loadings from all runs that used features from chromosome two. The color bars associated with the axes label the topic's strongest tissue association, if any. Left: correlations calculated using the residuals after regressing coded MAF out from the expectation of the genotype loadings. Right: correlations between the expected value of gene expression loadings.

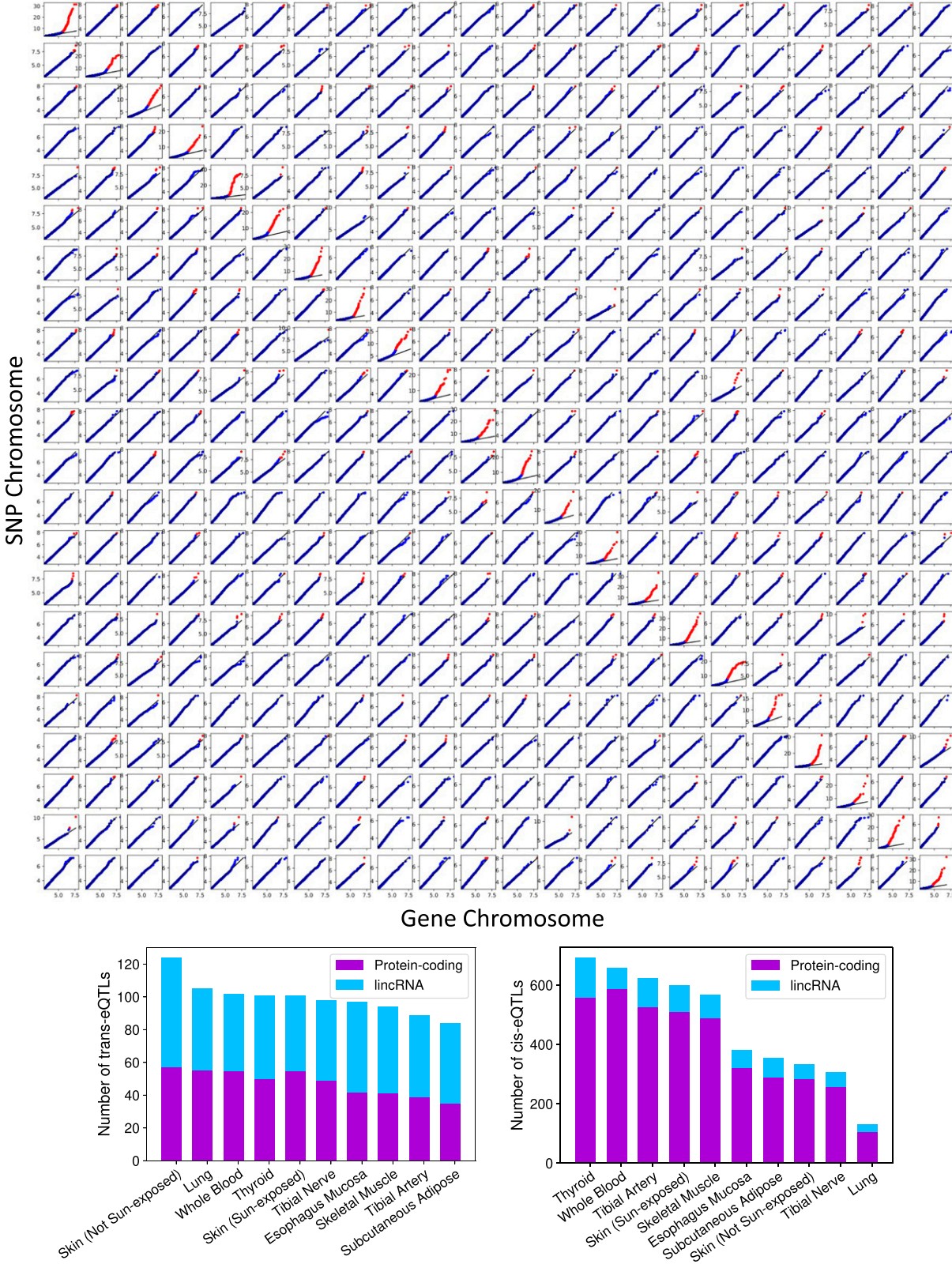

**Figure 4. Characterization of cis- and trans-expression quantitative trait loci (eQTLs) between top-ranked features in each factor.**
Top: for each of the 484 model runs, the ordered true MatrixEQTL association −log$_{10}$(P-values) (y-axis) are plotted against ordered −log$_{10}$(P-values) from tests using the same features but permuted expression and covariate data (x-axis). Clear cis-eQTL enrichment is present across all intrachromosomal runs. Points at which the ordered

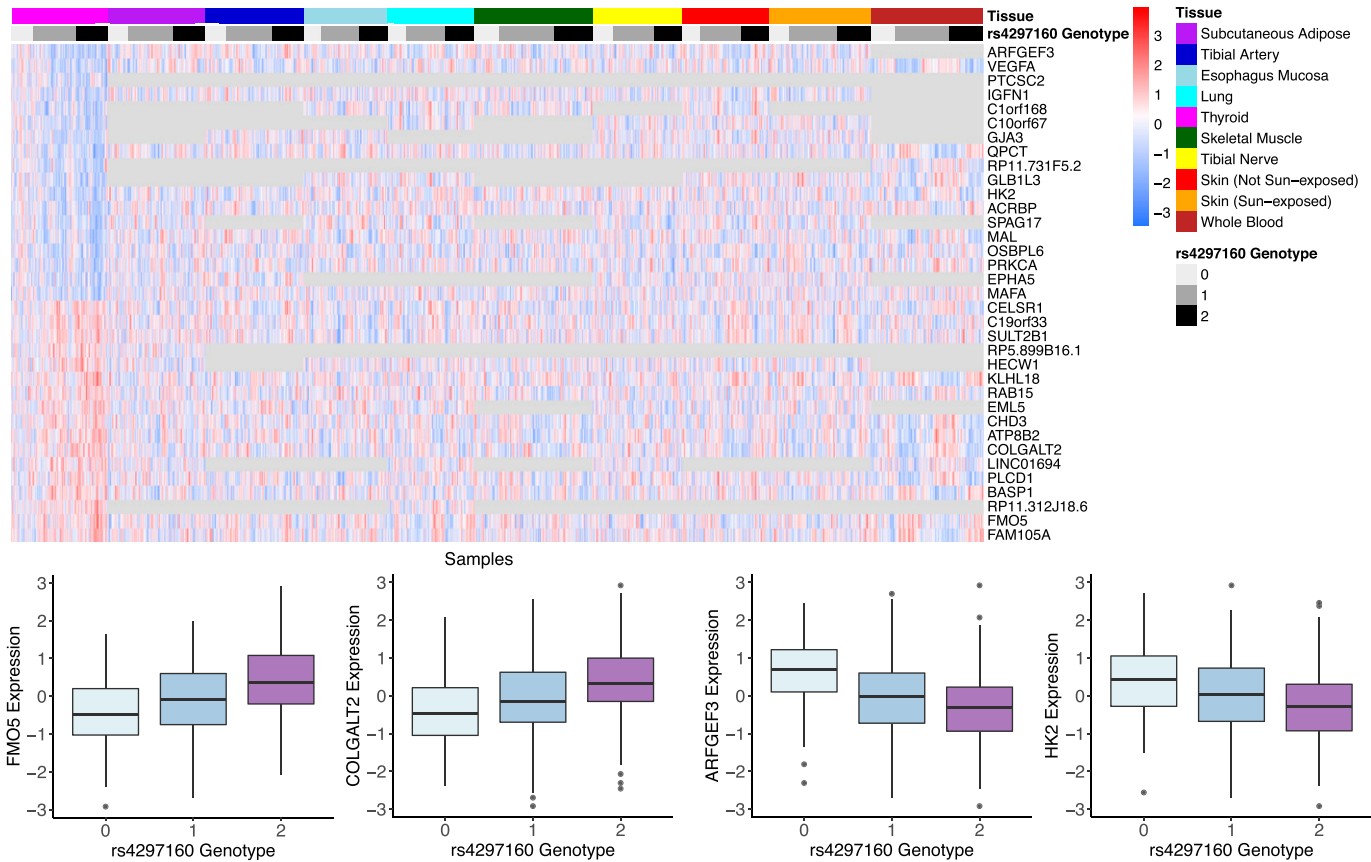

**Figure 5.  Exploring the 34 trans-eGenes associated with a single locus in thyroid.**
Top: heatmap of quantile-normalized expression values for the trans eGenes associated with rs4297160 along with *PTCSC2*. Bottom: visualization of four of the trans-expression quantitative trait loci.

factors for each tissue separately using MatrixEQTL (Shabalin, 2012) on held-out data (further referred to as the multivariate testing approach; Fig 4). Of 10,855,277 total tests, we found 4,645 cis-eQTLs at BH FDR < 0.1 across all 10 tissues including 2,149 unique eVariants and 1,868 unique eGenes (Table S4). Most of these cis-eQTLs (3,923) affect protein-coding genes, with a minority (722) acting on long intergenic noncoding RNA (lincRNA) genes (Fig 4). Thyroid had the most cis-eQTLs, 692, followed by whole blood with 657.

We also discovered 995 trans-eQTLs at BH FDR < 0.1, which include 901 unique trans-eGenes and 979 unique trans-eVariants (Table S4). In contrast to both the data, which consist of 86% protein-coding genes, and the cis-eQTLs, trans-eQTLs have an approximately equal number of lincRNA- and protein-coding eGenes (517 and 478, respectively; Fig 4). Although these numbers may seem surprising, lincRNAs localize to the nucleus and are chromatin-associated, often acting in trans through chromatin modifiers (Holdt et al, 2013; Hacisuleyman et al, 2014; Cao et al, 2021). However, trans-eQTLs found using bulk RNA-seq can appear because of sample cell type proportions (Võsa et al, 2021). Because

lincRNAs show more cell type–specific expression than protein-coding transcripts, this could contribute to the imbalanced numbers of noncoding and coding trans eGenes (Liu et al, 2016; Grassi et al, 2021). The skin (not sun-exposed) had the highest number of trans-eQTLs (124), followed by the lung with 105. Not surprisingly (GTEx Consortium, 2017), the cis-eQTL enrichment is much stronger and more consistent than the trans-eQTL enrichment (Fig 4). These eQTL mapping results highlight the associations between SNPs and genes loaded onto a common factor and suggest that traditional eQTL candidates may be identified using the TBLDA factors.

Next, we restricted our analysis within each tissue to the respective tissue's associated factors. We found 746 cis-eQTLs (618 unique eGenes and 662 unique eVariants) and 939 trans-eQTLs (853 unique trans-eGenes and 933 unique trans-eVariants) at BH FDR < 0.1 (Table S5). Whole blood had the most cis-eQTLs and the most trans-eQTLs. Similarly, to the full analysis above, the trans-eQTLs have an approximately equal split of lincRNA and protein-coding eGenes, whereas the cis-eGenes are mostly protein-coding (Fig S3). Of the discoveries, 877 (93.4%) of the trans-eQTLs and 324 (43.4%) of

true −log$_{10}$(P-value) is greater than the maximum permuted −log$_{10}$(P-value) are colored in red to highlight deviation. Bottom: histograms depicting the numbers of trans- (top) and cis- (bottom) eQTLs mapped per-tissue and split by gene type.

the cis-eQTLs were novel, meaning not below the significance threshold in the unrestricted multivariate test using all common factors. The fact that 933 (93.8%) of the trans-eQTLs found by the full multivariate test were not found in the tissue-associated factors indicates that most of the trans associations found by the model are not in specific tissue-factor pairs.

Thus, to increase power to find trans-eQTLs shared across tissues, we next limited association tests to features in general factors that were not linked to any tissue. This approach yielded 964 trans-eQTLs (863 unique trans-eGenes and 945 unique transeVariants) and 2,901 cis-eQTLs (1,210 unique eGenes and 1,355 unique eVariants; Fig S3 and Table S6). Here, the tibial nerve had the most trans-eQTLs (113), whereas the thyroid produced the most cis-eQTLs (415). The reduction in test numbers allowed 387 new trans-eQTLs and 327 cis-eQTLs to move below the significance threshold relative to the unrestricted multivariate test.

Inferred covariates such as PEER factors are known to capture and thus inadvertently control for broad regulatory effects that may have a true genetic basis, potentially removing broad trans-eQTL signals (Rakitsch & Stegle, 2016; GTEx Consortium, 2017). To test whether factors in our model find these kinds of regulatory hotspots, we ran the same eQTL mapping as before except excluding all PEER factors from the covariate matrix. This resulted in fewer total cis- and trans-eQTLs (2,456 and 882, respectively, at BH FDR < 0.1; Table S7). However, the proportion of unique eVariants to trans-eQTLs versus including PEER factors was lower (0.91 versus 0.98), suggesting that, to some extent, PEER factors do remove trans-acting pleiotropic signals that are captured by our model. In line with their supposed mechanisms of action, 69.1% (1,698) of these cis-eQTLs overlapped with our prior analysis controlling for PEER factors, whereas only 2.3% (20) of these trans-eQTLs were also found when controlling for PEER factors in the association analysis.

Next, we explored the overlap of our eQTLs and the GTEx consortium cis- and trans-eQTL list, produced by the consortium through an exhaustive tissue-specific testing approach (GTEx Consortium, 2020). A majority (4,645, 98.3%) of the multivariate TBLDA cis-eQTLs were in the GTEx cis-eQTL list. Of the multivariate trans-eQTLs, just one overlapped with the 2,629 genome-wide GTEx trans-eQTLs in the top 10 tissues. However, 26/438 (5.9%) of the GTEx trans-eQTLs in the relevant tissue, skin (sun-exposed), included that common eGene (*ALDH3B2*). Although we fail to capture this extended signal because we use an LD-pruned SNP set, our model still groups the gene together with its genomic hotspot. Furthermore, although the eVariant is not shared, 37/439 (8.4%) of the GTEx skeletal muscle transeQTLs include *RP11-65J3.3*, which we identify as a trans-eGene in that tissue. Taken together, these results suggest that our approach finds overlapping cis-eQTL signals but expands our ability to identify broad-acting trans-eQTLs in these bulk data.

One interesting example from the model fit using all GTEx samples is the trans eVariant rs4297160, which is associated with both *MAPRE3* (*P*-value $P \leq 6.6 \times 10^{-11}$) and *ARFGEF3* (*P*-value $P \leq 2.2 \times 10^{-16}$) in the thyroid and sits in the 9q22 locus (Fig 5). Specifically, rs4297160 is located within the lincRNA gene *PTCSC2*, which has been linked to a predisposition for papillary thyroid cancer (He et al, 2015). The 9q22 locus houses the thyroid-specific TF *FOXE1*, which shares a bidirectional promoter with *PTCSC2* (Wang et al, 2017).

Furthermore, the 9q22 locus was previously found associated in trans with *ARFGEF3* in the thyroid (GTEx Consortium, 2017). Notably, PEER factors were shown to capture and therefore control for broad regulatory signals from that locus (GTEx Consortium, 2017); in line with this, in association tests from the model trained on all GTEx data and run without PEER factors as covariates, rs4297160 was a trans-eVariant for 34 different genes in thyroid, including *HECW1* (regulates the degradation of thyroid transcription factor 1 [Liu et al, 2019]), *COLGALT2* (down-regulated in patients with thyroid orbitopathy [Khong et al, 2015]), and *FMO5* (expressed in endocrine cells that produce hormones that regulate metabolism [Xu, 2017]; Fig 5). These 34 trans-eGenes are enriched in the HIF-1 signaling KEGG pathway and two SP1 TF motifs, lending support for transcriptional co-regulation (g:Profiler [Raudvere et al, 2019] adjusted *P*-value < $1.1 \times 10^{-2}$ for all).

We evaluated the increase in statistical power compared with the univariate approach because of our reduced multiple testing burden. The cis-eQTL *P*-values with BH FDR < 0.1 from our method have a different distribution from the GTEx cis-eQTLs found via exhaustive search (Kolmogorov–Smirnov test statistic 0.11, *P*-value $P \leq 1.0 \times 10^{-16}$). Our cis-eQTLs found via TBLDA are a subset of all true associations, and they tend to have more moderate associations than the set of GTEx cis-eQTLs (Fig S4). We expect this because TBLDA factors identify associated groups eQTLs, each of which may only have a small univariate effect size. Moreover, because we are computing *P*-values on a small held-out sample, we cannot achieve the statistical significance for the same association test as that test applied to a larger sample.

## Discussion

In this paper, we present a probabilistic telescoping bimodal latent Dirichlet allocation (TBLDA) model that uncovers shared latent factors between bulk RNA-seq expression and genotype data when there is no one-to-one mapping among the samples for each data modality. The model takes raw counts as input, which avoids any potential data skewing because of normalization. We fit the model in an unsupervised manner using gene expression data from 10 tissues in the GTEx v8 release and matched donor genotypes. We intentionally exclude hard-coding tissue labels into the TBLDA because different tissues have a range of overlapping cell types, meaning that samples from certain tissues will share varying proportions of underlying processes and eQTLs. In addition, in large data sets, some samples may be mislabeled. Using known GTEx covariates, we established that the recovered topics reflected meaningful biology such as sample cell type proportion (Fig S2). Robust gene sets in tissue-associated factors were enriched for functionally relevant pathways (Table S1). Causal eVariants identified by our method are known to be enriched in a variety of genomic regulatory regions (Albert & Kruglyak, 2015); top-ranked robust tissue-associated SNP sets in our model were likewise enriched, demonstrating motifs of known eVariants (Table S2).

Running linear association tests on top-ranked features from each factor using MatrixEQTL (Shabalin, 2012) on a held-out test set,

we found 4,645 cis-eQTLs and 995 trans-eQTLs at BH FDR 0.1. By restricting association tests to the top features per factor in our model, we decrease the multiple testing burden and increase power for mapping trans-eQTLs on a small held-out test set. This is demonstrated by the fact that 994 of our trans-eQTLs were not identified in the exhaustive genome-wide GTEx analysis. A critical caveat of our approach is that, with a finite number of topics, we do not expect the model to capture all true eQTLs; however, we show that it does reproducibly identify novel and functionally relevant eQTLs. Taken together, these results demonstrate that our method successfully learns biologically meaningful shared topics across gene expression and genotype data. TBLDA is a natural framework to investigate cell type-specific eQTLs using single-cell RNA sequencing data, and we are currently exploring this promising future direction.

There are several potential points of contention in our model. First, although the model's probabilistic nature provides important measures of uncertainty for noisy genomic data, because of our inference procedure, the posterior should be interpreted with caution because variational inference is known to underestimate the posterior variance (Giordano & Broderick, 2015 *Preprint*). Second, because we do not include a private subspace for gene expression, true latent components that reflect expression-specific variation such as batch effects will be forced to contribute to the modality-shared factors. We believe this is important to retain signal for broad regulatory effects that especially affect trans-eQTL discovery. Nevertheless, if the model is used in a context such as single-cell RNA sequencing, where there are known and strong expression-specific covariates such as batch effects, this design choice should be reconsidered. Furthermore, a natural question that arises for all parametric latent factor models is how to determine the number of topics. We stress that there is no "correct" topic number and the user will want to make a reasonable trade-off between computational speed for inference and the granularity of signal captured. In practice, we recommend anywhere from 20 to 150 factors depending on the size of the data set. Given these qualities, natural extensions to the model include adding latent or semi-supervised expression-specific topics and extending it to a nonparametric framework.

## Materials and Methods

Given a genotype matrix and an RNA sequencing expression matrix, our goal is to find latent factors that capture groups of SNPs and genes that covary across samples. We have two input matrices: a RNA-seq count matrix $X \in R^{G \times L}$ for $G$ genes across $L$ samples and a genotype matrix $Y \in R^{S \times N}$ in minor allele dosage format (0,1,2) for $S$ SNPs across $N$ individuals. We henceforth refer to genes and SNPs as features. Each individual $i \in 1,..., N$ contributes at least one sample, and every sample $\ell \in 1,..., L$ comes from exactly one known individual; this is the telescoping property of the data.

We define $K$ latent topics where (i) each sample $\ell$ has topic membership proportion $\boldsymbol{\phi}_\ell \in S^K$ and (ii) each individual $i$ has topic membership proportion $\boldsymbol{\theta}_i \in S^K$, where $S^K$ denotes the K-dimensional simplex where all values are positive and sum to 1 (Figs 6 and

S5). Topics are modeled as distributions over features, where, similar to LDA, gene expression topics $\boldsymbol{\psi}_k \in S^G$ with $k = 1,..., K$ are located on the simplex (Blei et al, 2003). The expression portion of the model that describes gene probabilities $\boldsymbol{\pi}_\ell$ is:

$$\boldsymbol{\psi}_k \sim \text{Dirichlet}(\boldsymbol{\xi}) \tag{1}$$

$$\boldsymbol{\phi}_\ell \sim \text{Dirichlet}(\boldsymbol{\sigma}) \tag{2}$$

$$\boldsymbol{\pi}_\ell = \Psi\boldsymbol{\phi}_\ell \tag{3}$$

$$\mathbf{x}_\ell \sim \text{Multinomial}(C_\ell, \boldsymbol{\pi}_\ell) \tag{4}$$

where $C_\ell$ is the observed total read count in sample $\ell$ and the matrix $\Psi$ comprises the concatenated column vectors $\boldsymbol{\psi}_k$ with $k = 1,..., K$.

The sample to individual mapping is encoded in $\boldsymbol{\omega}_i \in \{0,1\}^L$, an indicator vector for individual $i$, where $\omega_{i\ell} = 1$ if sample $\ell$ originates from individual $i$. The "telescoping" portion of the model projects the shared factors in $\Phi$ (the matrix formed by concatenating all $\boldsymbol{\phi}_\ell$, $\ell = 1,..., L$) between the sample and individual spaces:

$$\boldsymbol{\theta}_i = \frac{1}{\sum_{\ell=1}^{L} \omega_{i\ell}} \Phi\boldsymbol{\omega}_i \tag{5}$$

In contrast to the expression topics $\boldsymbol{\psi}_k$, genotype topics are modeled independently over SNPs and consist of independent $\lambda_{jk}$ as in the structure model (Pritchard et al, 2000):

$$\lambda_{jk} \sim \text{Beta}(\zeta_j, \gamma_j) \tag{6}$$

We include a modality-specific (private) subspace for genotype to control for ancestral structure in mixed-population samples, which consists of nonnegative factor and loadings matrices $\beta$ and $\tau$ (see the Extended Methods section) given $Q$ ancestry factors. The weight of the private versus shared genotype subspaces is determined by $0 \le \alpha \le 1$ and learned during inference. The model does not include a gene expression-specific latent space to avoid losing any broad regulatory signal that is genotype-dependent (Rakitsch & Stegle, 2016), as is often the case with transeQTLs (GTEx Consortium, 2017). This final portion of the model, which covers minor allele probabilities $\rho_{ij}$, is as follows:

$$\alpha \sim \text{Uniform}(\delta, \mu) \tag{7}$$

$$\rho_{ij} = \alpha \sum_{q=1}^{Q} \beta_{jq}\tau_{qi} + (1-\alpha)\left(\sum_{k=1}^{K} \lambda_{jk}\theta_{ki}\right) \tag{8}$$

$$y_{ij} \sim \text{Binomial}(2, \rho_{ij}) \tag{9}$$

We use stochastic variational inference to compute posterior estimates for $\Phi$, $\Lambda$, $\Psi$, and $\alpha$ (see the Extended Methods section for details).

In this multimodal version of LDA, the same latent factors are shared across data modalities, allowing features of each modality to be directly linked together. Critically, because our framework directly models count data, we avoid spurious or distorted signals

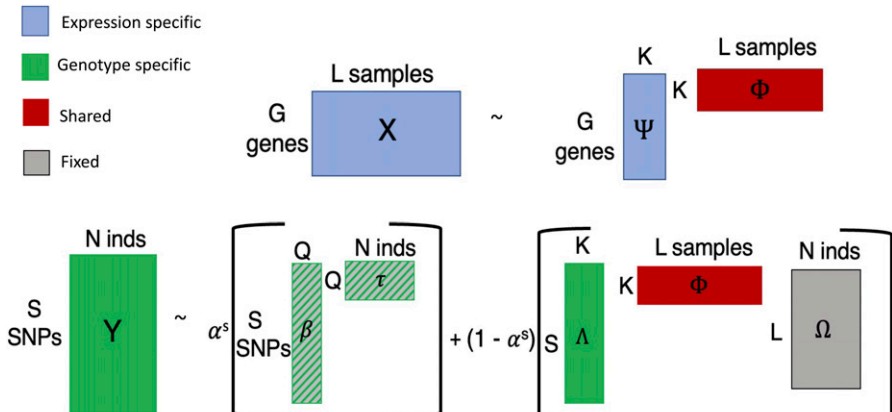

**Figure 6. Model visualization.**
Explicit representation of the model described in Materials and Methods section Equations (1)–(9) with dimensions drawn out. Each portion of the model is color-coded according to modality. Gray represents the known mapping between samples and individuals. The ancestry portion is striped because it is learned before fitting the shared model portion.

through data normalization (Robinson et al, 2010; Love et al, 2014; Hicks et al, 2018), and because of the nonnegative factors, the components capture parts-based patterns instead of global patterns (Lee & Seung, 1999; Townes & Engelhardt, 2021 Preprint). The multinomial distribution allows us to separate out variation because of library-size effects from the underlying compositional variation, which is more biologically relevant.

Features that have higher weights within a topic ($\psi_k$, $\lambda_{jk}$) have a larger relative contribution. However, the proportion of total counts for each gene varies widely. Genes with higher counts may dominate certain topics merely because of their high expression levels, overshadowing lower expressed genes that are actually more informative for that topic compared with others. Consequently, instead of using the raw expected loadings, we determine the importance of each feature across topics by ranking the average 2-Wasserstein distance between the posterior variational distributions. This allows us to control for both average feature counts and varying uncertainty in model estimation by using the full information provided in the posterior estimates. In particular, the 2-Wasserstein value, also known as the "Earth-movers distance," is specifically designed as a distance metric between two empirical densities, unlike KL divergence, which is not a symmetric measure and often performs poorly when the two densities are defined on unequal basis measures. SNP minor allele frequency (MAF) is much less variable than gene total counts. To control for allele counts, we rank SNPs after regressing out the coded MAF in each loading (see the Extended Methods section for details).

## Extended methods

### Feature selection
After Jo et al (2016) Preprint, we used plink 1.9 (Purcell et al, 2007) to trim the GTEx v8 whole-genome sequencing SNP sets such that no two SNPs within a 200-Kb window have a Pearson correlation ≥0.2. SNPs with imputed genotypes were removed, yielding 202,111 remaining SNPs across 831 individuals. All gencode v26 autosomal lincRNA and protein-coding genes from the 5,781 samples with genotypes were considered. We retained the 19,534 autosomal genes with a median RNASeQC v1.1.9 (Graubert et al, 2021) read

count of at least five in at least one tissue. SNPs and genes were split into 22 groups by chromosome.

### Ancestry structure
We ran terastructure (Gopalan et al, 2016) on the 202,111 LD-trimmed SNPs with the following options: -rfreq = 40,222 and -K = 5. The resulting allele frequencies (beta.txt) and admixture proportions (theta.txt) output matrices were assigned to $\beta$ and $\tau$ (Equation (8)) to produce the genotype-specific portion of the model.

### TBLDA model runs
We used Pyro v1.4.0 Bingham et al's (2019) stochastic variational inference framework to fit the model, using pyro.poutine.scale (scale = $1.0 \times 10^{-6}$) for numerical stability, an Adam optimizer, and a learning rate of 0.05. $\xi$ and $\sigma$ were set to symmetric one vectors, $\zeta_j$ = $\gamma_j$ = 1, $\delta$ to 0.05, and $\mu$ to 0.85. The model was fit separately for feature sets from each chromosome combination, for a total of 22 × 22 = 484 runs. Let $x$ be the average ELBO over the latest 1,000 epochs and $y$ be the average ELBO over the 1,000 epochs before those. Runs were terminated when $\frac{y-x}{y} \leq 1 \times 10^{-4}$. The model run for chromosome 12 with 9,819 SNPs across 831 individuals and 1,124 genes across 5,781 samples required 3G and converged in 17 h using four CPUs. Code to run TBLDA is available at https://github.com/gewirtz/TBLDA.

### Insights on setting hyperparameters
(i) $\sigma$: This is the hyperparameter for the sample-topic proportion $\phi_\ell$. This vector should be kept symmetrical, and we recommend running the model with an uninformative prior where $\sigma$ = 1. If the user wants to fit a model where samples comprise many topics, they should running the model with a more concentrated prior, setting $\sigma > 1$. Conversely, if the user wants each sample to be drawn from only a few topics, they can set a sparser prior where $\sigma < 1$.

(ii) $\zeta, \gamma$: These are the hyperparameters for the SNP loadings $\lambda_{jk}$. Although we set them equal to one for an uninformative prior, users could also set them to be less than one for a sparsity-inducing prior. In that case, a common choice would be setting the hyperparameters to 1/$K$, where $K$ is the number of topics. We recommend trying an uninformative prior first to let the

data lead, and if topic collapse is observed, next try the sparsity-inducing prior.

(iii) $\xi$: This is the hyperparameter for the gene loadings $\phi_k$. We follow the same guidance as for $\zeta$ and $\gamma$, except that $\xi$ is the parameter vector for the Dirichlet, which generalizes the beta.

(iv) $\mu, \delta$: These are the hyperparameters for $\alpha$, which controls the mixture proportion for the genotype-specific versus shared space. The model will naturally put more weight on the separate genotype portion, and we want to restrict this so that TBLDA learns more shared structure. Thus, we want to restrain $\alpha$ from getting too close to zero or one. In practice, we recommend setting $\mu \geq 0.1$ and $\delta \leq 0.8$.

**Running TBLDA on samples with additional technical covariates** As a general practice for downstream model analysis, we recommend that users identify factors associated with all covariates (e.g., following the described methodology for tissue and cell type enrichment score associations). For example, when given technical covariates such as batch, users may remove all batch-associated factors from downstream analysis.

### Feature ranking

After regressing out allele frequency, we take the top 10% of SNPs from each loading with the highest absolute value residuals. The 10% of genes from each loading with the highest 2-Wassterstein distances are considered the top gene features. Because the feature numbers vary by chromosome, runs have differing numbers of top features associated with their factors.

### Functional enrichment data

The tissue-specific TF list originated from Table S3 in Sonawane et al (2017). To conduct GSEA, we used all biological process terms from GO v6.2 that had at least three genes in common with our analysis feature set. We used LDSC's baselineLD v2.1 (Bulik-Sullivan et al, 2015) genome annotations to compute SNP set enrichments. We did not consider MAF bin classes.

### Tissue-associated genes

For each tissue, the set of robust tissue-associated genes consists of the genes that are top-ranked in at least one tissue-associated factor across all TBLDA runs.

### Tissue-associated SNPs

For each tissue, we calculated the 75th percentile of the distribution of total tissue-associated factors across all runs that each top-ranked SNP is associated with. SNPs that are top-ranked in at least the 75th percentile of each tissue's associated factors across all runs comprise the set of robust tissue-associated SNPs.

### eQTL pipeline

We held out two randomly selected samples from all individuals who contributed four or more samples to use for the eQTL pipeline and used all remaining samples to fit TBLDA. We used MatrixEQTL v2.3 (Shabalin, 2012) with modelLINEAR to run the eQTL testing. Expression for all genes that passed a 0.8 mappability filter was quantile-normalized as input. Sex, PCR, platform, the top five genotype principal components, and the top 60 PEER factors per

tissue were included as covariates. FDR was computed using the Benjamini–Hochberg procedure over each run for protein-coding and lincRNA genes separately. We note that the 0.1 FDR we use is more lenient than the 0.05 FDR threshold used in the v6p GTEx paper (GTEx Consortium, 2020), although the v8 GTEx trans-eQTLs we compare to were also identified using a 0.1 FDR threshold.

### Robust components

We computed the correlation of each factor loading with all other loadings from runs on the same chromosome. Any factor with more than two loading Kendall $\tau > 0.15$ for SNPs and three Pearson $r^2 > 0.95$ for genes was flagged—along with the highly correlated factors—as a robust component. For each robust component, we averaged the constituent loadings to produce a representative factor loading. All components whose representative loadings exceeded $r^2 > 0.95$ were further collapsed into a single robust component.

### Data access

All raw sequencing and genotype data from GTEx v8 used in this study can be found in dbGaP under accession number phs000424.v8.p2.

# Supplementary Information

# Acknowledgements

ADH Gewirtz, FW Townes, and BE Engelhardt were funded by Helmsley Trust grant AWD1006624, NIH NCI 5U2CCA233195, NIH NHLBI R01 HL133218, and NSF CAREER AWD1005627.

### Author Contributions

ADH Gewirtz: conceptualization, data curation, software, formal analysis, validation, investigation, visualization, methodology, and writing—original draft, review, and editing.
FW Townes: conceptualization, supervision, and methodology.
BE Engelhardt: conceptualization, resources, supervision, funding acquisition, methodology, project administration, and writing—original draft, review, and editing.

### Conflict of Interest Statement

BE Engelhardt is on the SAB of Creyon Bio, ArrePath, and Freenome.

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
