## [Reviewer comments · Life Science Alliance]

Life Science Alliance

Telescoping bimodal latent Dirichlet allocation to identify expression QTLs across tissues

Ariel Gewirtz, F Townes, and Barbara Engelhardt

DOI: <https://doi.org/10.26508/lsa.202101297>

Corresponding author(s): Barbara Engelhardt, Gladstone Institutes

Review Timeline:

Submission Date:	2021-11-11
Editorial Decision:	2022-01-11
Revision Received:	2022-06-07
Editorial Decision:	2022-07-08
Revision Received:	2022-07-15
Accepted:	2022-07-18

Scientific Editor: Novella Guidi

Transaction Report:

January 11, 2022

Re: Life Science Alliance manuscript #LSA-2021-01297-T

Barbara E Engelhardt
Princeton University, Gladstone Institutes

Dear Dr. Engelhardt,

Thank you for submitting your manuscript entitled "Telescoping bimodal latent Dirichlet allocation to identify expression QTLs across tissues" to Life Science Alliance. The manuscript was assessed by expert reviewers, whose comments are appended to this letter. We, thus, encourage you to submit a revised version of the manuscript back to LSA that responds to all of the reviewers' points.

Thank you for this interesting contribution to Life Science Alliance. We are looking forward to receiving your revised manuscript.

Sincerely,

B. MANUSCRIPT ORGANIZATION AND FORMATTING:

Reviewer #1 (Comments to the Authors (Required)):

In this paper, the Authors propose a telescoping bimodal latent Dirichlet allocation (TBLDA) framework for gene expression analysis. The core of the method is a latent variable model that allows sharing topics across transcript levels and genotypes. Amongst other features, this allows multiple RNA-sequencing samples to correspond to a single individual's genotype, and using a natural generative model for counts, additional normalization is not needed before application. As a result, the model captures ancestral structure, as well as biological signal; it includes pure factor analysis and independently modeling expression and genotype structure as special cases, thus generalizing these commonly used models. The model estimates can be used for mapping, and the Authors find many associations there.

I have major and minor comments which should only require limited computational experiments and additional discussion, but the paper is very nicely presented overall. It is an interesting method that deserves to be shared with the community.

Major comments

* Model and testing

- The proposed model explains structured variation in both SNP and gene expression data. Performing a statistical test on top effectively results in double dipping ("To test whether traditional eQTLs ascertained using univariate tests are captured by TBLDA, we ran a linear model for association between the top 10% most informative SNPs and genes on common factors"), a bit like first running a loose threshold eQTL mapping pipeline, and then using a stringent cutoff for testing - the criterion for being tested is to have signal. We may have misunderstood an aspect of methods that makes this criticism invalid. If this is the case, please clarify how double dipping is avoided; if not, please temper claims about improvements in power, and rather present the trans results especially as a useful way to explore the genetic associations.

- Applying such models in practice requires setting priors. Usually, most of their influence is overshadowed by data, but there are a few key ones whose settings substantially impact the output (e.g. we suspect δ and μ in this case to control the mixture proportion range). Please elaborate [either with simple experiments or analytical estimates] the regimes in which people should be taking care with the prior when applying the models. E.g. is it bad to allow δ and μ close to 0/1; are the Beta priors for SNP factor loadings important to keep flat (ϵ , σ of 1), U-shaped, or concave? The extra guidance would help people apply the model.

* eQTL mapping and interpretation

- For the eQTL mapping, the increase in the number of trans-eQTL discovered and highlighted in the discussion cannot be solely attributed to the model. As you point out, in comparison to the the GTEx trans-eQTL mapping the number of tests conducted is far lower (10,855,277 versus 5×10^{11} tests per tissue, GTEx consortium, 2020 supplementary materials p. 20) due to selecting the top 10% most informative SNPs and genes. Additionally, the FDR cutoff used by the authors is higher than used in the 2020 GTEx paper (0.1 versus 0.05 and 0.01). Therefore, the increased discovery in trans-eQTL could be attributed to the reduced multiple testing burden as well as the less stringent FDR cutoff. To clarify the claim about model's benefit, please match the FDR, and ideally, the test count.

- The evidence presented to support a trans effect of rs4297160 regulating 34 genes should be strengthened. A gene set enrichment analysis of the 34 trans eGenes would be informative to add evidence to transcriptional co-regulation. Transcription factor binding site analysis of the 34 eGenes could support a regulatory role of the FOXE1 transcription factor. Additionally, functional evidence linking PTSC2 to the 34 eGenes would also add evidence of a trans mechanism.

- When conducting the enrichment of transcription factors in different tissues, please check how many of the tissue-specific transcription factors are found in a tissue gene set where they should not be found, to evaluate a biological false positive rate.

- If we understood correctly, the number of cell-types tested is only 5 from 8 tissues (11 tissue and cell-type pairs). In order to conclude that the model effectively captures sample cell-type composition more cell-types should be tested if the data are available.

Minor comments [no response required, only edits if the Authors decide so]

- The word "telescoping" is used in title, abstract, intro, and discussion, but never defined or explained.
- The model "avoids normalization" is a bit strong, as a total gene count parameter is fit, which implicitly acts as normalization.
- It was a little annoying to have ϕ not visibly bold to indicate it's a vector. It's understandable that it does not strictly fit in a plate as the variables are not conditionally IID, but it did confuse our readers to not easily understand that it also has the dimensionality of the latent space.
- In Figure 4, it is difficult to match the colors in the cluster map to the colors in the legend. Is it true that 'Skin (Sun-exposed)' and 'Skin (Not Sun-exposed)' are collapsed in Figure 4?
- While the code is available online, we recommend that the authors include a reference to the github in the manuscript.
- The authors state "Taken together, the two skin tissues had the largest group with 6,515 genes, while whole blood had the largest number of unique genes considering the other sets (1,983; Fig. 2)". I cannot find the gene-factor information in Figure 2 that is relevant to this statement. Could the authors elaborate in an appropriate place in text?
- Given that most of the thresholds in the right side of Figure 3 are not very strict, I'm wondering how strong the conclusions drawn from this figure are. Could the authors provide additional evidence that these regions 'make sense' (this might be most interesting for chromosome 4 and subcutaneous adipose, since this is the strongest suggested link)?
- The left side of Figure 3 is never referred to even though the relevant text is there. Could the authors please indicate in the text where the figure becomes relevant? This would improve readability.
- "clusters of co-regulated genes and SNPs" - it's not clear what it means for a SNP to be regulated.
- The 2-Wasserstein distance for comparing variational posteriors is an inspired, but surprising choice. Could the Authors elaborate (and perhaps evaluate on a small example - but citation enough if there is a clear one) what its impact is compared to more commonly used metrics, such as KL divergence?

Reviewer #2 (Comments to the Authors (Required)):

Review: Telescoping bimodal latent Dirichlet allocation to identify expression QTLs across tissues

The authors propose a probabilistic model to identify latent topics jointly from genotype and gene expression data across multiple tissues. The estimated topics can be interpreted biologically and, in particular, they can be used to preselect genetic variant - gene pairs for eQTL testing.

The mathematical model underlying the method is carefully designed and the application on the GTEX v8 data looks convincing, as they detect many novel associations.

However, the manuscript itself could be improved, including a better description of the methods at multiple points and easier access for interested user without a strong statistical background. What remains unclear in the manuscript, is how simple an application would be for a potential user, regarding availability of code and estimated runtime.

Major Comments:

1. Making the code of the method available is crucial to enable other researchers to use their method.
2. The method is applied separately for each chromosome, both for genes and SNPs. Why did the authors choose this strategy? This will weaken the approach, as it cannot capture relationships across genes on different chromosomes and the same for SNPs on different chromosomes. If the reason is runtime or memory restriction, this should be stated in the manuscript. In general, reporting the runtime and memory requirements are very valuable information for potential users.
3. The authors state that they don't want to model a gene expression-specific latent space in the fear of removing broad biological signal. But later they include PEER factors in their matrix eQTL analysis. Is that not contradicting with their initial reasoning? And if they want to include gene specific latent factor, it would be much more elegant to model them directly over their TBLDA model instead of using PEER afterwards (also easier for the user to not apply to many different methods for the eQTL analysis).
4. Linked to the last comment, the authors estimate the ancestry structure using their TBLDA model, but later for their matrix eQTL analysis, they do not include their own Ancestry Structure, but instead the top five genotype principal components. Why? Again, it would be nicer if the results from their model could be incorporated directly in the downstream eQTL analysis. In addition the number of eQTL discoveries for both approaches should be compared to each other (TBLDA genotype factors vs genotype PCs).
5. The authors state that in the discussion that single-cell RNA-seq data would require a correction for batch effects, using a private subspace for gene expression data. But also bulk data can suffer from batch effects. Is there a way to apply their TBLDA model on bulk data with batch effects or will it only work on data without batch effects (which would be quite a limitation)?
6. The method for selecting the top "features" is not explained well (p.9, Feature Ranking): they first explain how they select the top 10% features and then the top 10% genes? Is the first part with 10% features including only SNPs? If yes, is there are way

to determine the relative contribution of genetics and genes combined for one topic (so is the influence of genetics or gene expression stronger)? Please clarify notation and naming to avoid confusion.

7. Linked to the last comment, "tissue-associated SNPs" (p.9) are analyzed, but not tissue-associated genes, why?

8. Regarding the analysis of "robust components": for my understanding, robust components identify genes that are independent of association with SNPs and capturing often general tissue structure. Is this part of the model not hindering the identification of candidate eQTLs? Are the robust components somehow excluded when selecting candidate eQTLs?

9. The authors run first an eQTL analysis on the common factors. In contrast to the second step, the eQTL analysis on tissue-associated factors, they report here only the numbers without assessing their results compared to a full eQTL analysis. Do the eQTLs from the common factor approach also contain many novel eQTLs (as for the eQTLs from the tissue-associated factors)? The authors should show here if also the eQTLs analysis for common factors is a valuable selection strategy or if only the focus on tissue-associated factors brings new eQTL results?

10. The relation between cell type and LDA factors is very interesting. The authors should further explore if eQTL associated with these factors are enriched for cell type specific eQTL. This could be done exemplarily for whole blood, where both single cell (PBMC) and sorted bulk eQTL data sets (blueprint and immunexut) are available.

Minor Comments:

1. The model description is very mathematical, making it less understandable for readers without profound statistical background. A more abstract description of the model combined with a better visualization than Figure 1 would improve the manuscript. Supplementary Figure 1 is already better showing the connection between the parameters (better than Main Figure 1).

2. The authors report the known ancestry structure is not associated with shared factors (using correlation coefficient), but they do not report if the genotype-specific factors are correlated with the ancestry structure. Reporting for this some correlation value or similar would be valuable to prove that their genotype-specific subspace is indeed capturing the ancestry information.

3. Why was a cutoff of 40 chosen to define a tissue-specific factor?

4. Figure 5 is very difficult to read with all the tiny panels, the Figure 5 would benefit from a better visualization.

5. The manuscript contains a section "Methods" and another section "Extended Methods". At multiple points, it says in the text "see Methods" while "see Extended Methods" would be correct.

6. Was the LDSC annotation matched to the respective tissue? In other words were the SNPs indeed eQTL in the same tissue from which the enhancer annotations were obtained?

Reviewer #3 (Comments to the Authors (Required)):

The authors are presenting an approach for eQTL mapping exploiting the fact that multiple expression measurements might exist per subject. Whereas multiple eQTL mapping approaches addressing this issue have been proposed, the authors here present a method that works on raw read counts assuming proper (discrete) distributions of the data.

I do see a need for such an eQTL mapping method.

However, I found the presentation of the method exceptionally hard to follow. I had asked two of my team members to join me in reading and discussing the manuscript. Therefore, I am going to use 'we' subsequently, because everything that follows is the result of intense discussions among the three of us.

We have a range of major concerns that we couldn't resolve despite our discussion and hence, we did not go beyond discussing the model itself and did not evaluate the results. We feel the manuscript has to be majorly re-written to make it much more accessible to scientists not working with LDA on a daily basis. All the following points refer to the presentation of the model, mostly equations 1-9, Figure 1 and in part also the Extended Methods at the end.

1. Information on the tissue the sample comes from does not seem to be used. We feel this is valuable information left out of the mapping. We fear that the algorithm may learn tissue context as topics (one topic per tissue), which would be an unnecessary learning step, 'wasting' degrees of freedom and potentially adding additional noise (e.g. if samples are assigned to the wrong 'tissue topic'). At least, this point should be addressed in the manuscript.

2. The notation and indexing used in equations 1-9 is overly complicated. This is the biggest issue of the current manuscript. For

example, samples are indexed with l and with i . Using l each sample is already uniquely indexed and hence the second index is redundant. Why this double indexing? This is just creating problems for understanding the equations.

3. Further, the notation between expression-related and SNP-related symbols is inconsistent. For example, the expression matrix is denoted with X and the genotype matrix is denoted with Y . Thus, different symbols are used to distinguish genotype and expression. However, the dimensionalities of the matrices are denoted with the same symbol (F) but different superscripts (F^g and F^s). This is creating a lot of downstream issues. For example, we were wasting a lot of time trying to understand equation (4). It was clear that equation (4) would compute a vector over genes. Yet, the superscript 'g' in x^g fooled us, because we thought this would be the expression level of a particular gene 'g' until we realized this was just indicating that x would be a vector of gene expression. Yet, in this case the superscript is not needed, because X was already introduced as the matrix of gene expression values. And so on

4. The model plate diagram (Fig. 1) only partially helps, because it contains symbols that are not shown in equations 1-9. One has to refer to the Extended Methods to fully understand it.

5. We were discussing for a long time how and where correction for library size (read counts) is done, until we realized it is C_l , which is incorrectly denoted 'gene count' (should be 'read count'). Such mistakes further complicate the understanding.

6. Equation 3 seems to average the expression level of a gene over the topics. Correct? Why can this be done? Averaging over multiple topic assignments for a single observation may lead to infeasible averages. I.e. whereas either one or the other topic may be possible, what about the average?

7. Equation 7: is α user-defined or learned from the data? How is it set?

8. Equation 8 needs better explanation. For example, what does the K with superscript s mean?

9. "We include a modality-specific (private) subspace ..." Where is the product $\beta * \zeta$ coming from? What does it mean?

10. Extended Methods: For example in Ancestry Structure it says: "The resulting β and θ output matrices were assigned ..." or in Model Runs: " ξ and σ_i were set to one, δ to 0.05, ...". In all of those cases it isn't clear what these parameters and output matrices are. One would have to read all the cited papers to understand their meaning, whereas the Methods should be self-explanatory.

11. Stochastic variational inference: How would different parameters affect the results? Is there some motivation for the parameter choices? Are they simply the default values?

Response to Reviewer Comments

We thank the reviewers for their thoughtful comments. We respond to them here point-by-point, with the review comments in black text and our responses in blue italic text. We also highlight the manuscript changes in red; we note that we made substantial structural changes to the presentation of the model and the eQTL analyses following reviewer comments, but we did not make those full sections red despite the changes.

Reviewer #1 (Comments to the Authors (Required)):

In this paper, the Authors propose a telescoping bimodal latent Dirichlet allocation (TBLDA) framework for gene expression analysis. The core of the method is a latent variable model that allows sharing topics across transcript levels _and_ genotypes. Amongst other features, this allows multiple RNA-sequencing samples to correspond to a single individual's genotype, and using a natural generative model for counts, additional normalization is not needed before application. As a result, the model captures ancestral structure, as well as biological signal; it includes pure factor analysis and independently modeling expression and genotype structure as special cases, thus generalizing these commonly used models. The model estimates can be used for mapping, and the Authors find many associations there.

I have major and minor comments which should only require limited computational experiments and additional discussion, but the paper is very nicely presented overall. It is an interesting method that deserves to be shared with the community.

Major comments

* Model and testing

- The proposed model explains structured variation in both SNP and gene expression data. Performing a statistical test on top effectively results in double dipping ("To test whether traditional eQTLs ascertained using univariate tests are captured by TBLDA, we ran a linear model for association between the top 10% most informative SNPs and genes on common factors"), a bit like first running a loose threshold eQTL mapping pipeline, and then using a stringent cutoff for testing - the criterion for being tested is to have signal. We may have misunderstood an aspect of methods that makes this criticism invalid. If this is the case, please clarify how double dipping is avoided; if not, please temper claims about improvements in power, and rather present the trans results especially as a useful way to explore the genetic associations.

Thank you for this comment – we agree about the double dipping concern, and in our revision we have addressed this by holding out two tissue samples from every individual with four or

more tissue samples at random in order to allow associations to be tested on held out samples.

- Applying such models in practice requires setting priors. Usually, most of their influence is overshadowed by data, but there are a few key ones whose settings substantially impact the output (e.g. we suspect δ and μ in this case to control the mixture proportion range). Please elaborate [either with simple experiments or analytical estimates] the regimes in which people should be taking care with the prior when applying the models. E.g. is it bad to allow δ and μ close to 0/1; are the Beta priors for SNP factor loadings important to keep flat (eps, sigma of 1), U-shaped, or concave? The extra guidance would help people apply the model.

This is an important clarification. To address this concern about how to practically set the hyperparameters, we wrote a set of “rules of thumb” for different scenarios, included in the revised text as Supplemental File 1.

* eQTL mapping and interpretation

- For the eQTL mapping, the increase in the number of trans-eQTL discovered and highlighted in the discussion cannot be solely attributed to the model. As you point out, in comparison to the the GTEx trans-eQTL mapping the number of tests conducted is far lower (10,855,277 versus 5×10^{11} tests per tissue, GTEx consortium, 2020 supplementary materials p. 20) due to selecting the top 10% most informative SNPs and genes. Additionally, the FDR cutoff used by the authors is higher than used in the 2020 GTEx paper (0.1 versus 0.05 and 0.01). Therefore, the increased discovery in trans-eQTL could be attributed to the reduced multiple testing burden as well as the less stringent FDR cutoff. To clarify the claim about model's benefit, please match the FDR, and ideally, the test count.

This is a good observation. We rephrased the model and results comparison to incorporate not just the model and the number of tests conducted but also differences in thresholds. We would perform a quantile-quantile analysis with our trans-eQTL findings from the TBLDA model but there is insufficient overlap with the trans-eQTLs from GTEx to allow such an analysis. However, this implies that many of our findings from our model would not be found in the GTEx tests. We also revised our testing mechanism (as discussed above) to use held-out samples to avoid a possible double dipping problem.

It is true that the GTEx 2020 paper used a threshold of FDR > 5% for trans-eQTL discovery. We were the primary analysis group for trans-eQTLs, and we suggested for the main GTEx paper that 5% was prohibitively low as a threshold for distal associations because of the harsh burden of multiple hypothesis testing, as evidenced by the total of 143 trans-eQTLs across all tissue types. While we were overruled in the main GTEx paper, we will shortly preprint the

definitive trans-eQTL GTEx paper that conducts this same analysis but with a more reasonable 10% FDR threshold.

- The evidence presented to support a trans effect of rs4297160 regulating 34 genes should be strengthened. A gene set enrichment analysis of the 34 trans eGenes would be informative to add evidence to transcriptional co-regulation. Transcription factor binding site analysis of the 34 eGenes could support a regulatory role of the FOXE1 transcription factor. Additionally, functional evidence linking PTCSC2 to the 34 eGenes would also add evidence of a trans mechanism.

We agree that a stronger mechanistic story about PTCSC2 would be more compelling. We performed a GSEA and TFBS analysis with the 34 trans-eGenes and found enrichment in the HIF-1 signaling pathway and two SP1 transcription factor motifs; we have added this to the results section. We also looked for co-expression of PTCSC2 and the 34 trans-eGenes in heatmaps across tissues and added this to a revised Figure 6.

- When conducting the enrichment of transcription factors in different tissues, please check how many of the tissue-specific transcription factors are found in a tissue gene set where they should not be found, to evaluate a biological false positive rate.

We conducted this analysis and added the result to the revised manuscript.

- If we understood correctly, the number of cell-types tested is only 5 from 8 tissues (11 tissue and cell-type pairs). In order to conclude that the model effectively captures sample cell-type composition more cell-types should be tested if the data are available.

In our manuscript, we used all of the cell types available from the current data to construct our training data set. We are currently extending these ideas to single cell data without restrictions on cell types and will leave those results to the follow-up manuscript.

Minor comments [no response required, only edits if the Authors decide so]

- The word "telescoping" is used in title, abstract, intro, and discussion, but never defined or explained.

Thank you for pointing this out—we have defined telescoping better in the abstract and methods sections.

- The model "avoids normalization" is a bit strong, as a total gene count parameter is fit, which implicitly acts as normalization.

The total read count parameter is not fit – it is set using the input sample data. We have clarified this in the revised text.

- It was a little annoying to have ϕ not visibly bold to indicate it's a vector. It's understandable that it does not strictly fit in a plate as the variables are not conditionally IID, but it did confuse our readers to not easily understand that it also has the dimensionality of the latent space.

We have clarified the dimensionality of variables where they are introduced and revised the Methods section to make it more clear.

- In Figure 4, it is difficult to match the colors in the cluster map to the colors in the legend. Is it true that 'Skin (Sun-exposed)' and 'Skin (Not Sun-exposed)' are collapsed in Figure 4?

The two skin sample types are not collapsed – the colors are red (not sun exposed) and orange (sun exposed). We have kept the colors and tissue labels consistent across the different figures with the small exception of the factor labels in Fig. 2.

- While the code is available online, we recommend that the authors include a reference to the github in the manuscript.

We have added our github link to the abstract.

- The authors state "Taken together, the two skin tissues had the largest group with 6,515 genes, while whole blood had the largest number of unique genes considering the other sets (1,983; Fig. 2)". I cannot find the gene-factor information in Figure 2 that is relevant to this statement. Could the authors elaborate in an appropriate place in text?

We have removed the reference to Fig. 2 here.

- Given that most of the thresholds in the right side of Figure 3 are not very strict, I'm wondering how strong the conclusions drawn from this figure are. Could the authors provide additional evidence that these regions 'make sense' (this might be most interesting for chromosome 4 and subcutaneous adipose, since this is the strongest suggested link)?

Thank you for pointing this out – upon checking, we had previously plotted the p-values after multiple hypothesis correction and have updated the revised figure, which now shows the stronger enrichments.

- The left side of Figure 3 is never referred to even though the relevant text is there. Could the authors please indicate in the text where the figure becomes relevant? This would improve readability.

We have now referenced the left side of Figure 3 in the appropriate location of the revised text.

- "clusters of co-regulated genes and SNPs" - it's not clear what it means for a SNP to be regulated.

We have changed this to "covarying."

- The 2-Wasserstein distance for comparing variational posteriors is an inspired, but surprising choice. Could the Authors elaborate (and perhaps evaluate on a small example - but citation enough if there is a clear one) what its impact is compared to more commonly used metrics, such as KL divergence?

We have added the following explanation of the 2-Wasserstein distance to the revised text: "In particular, the 2-Wasserstein value, also known as the 'Earth-movers distance,' is specifically designed as a distance metric between two empirical densities, unlike KL divergence, which is not a symmetric measure and often performs poorly when the two densities are defined on unequal basis measures".

Reviewer #2 (Comments to the Authors (Required)):

Review: Telescoping bimodal latent Dirichlet allocation to identify expression QTLs across tissues

The authors propose a probabilistic model to identify latent topics jointly from genotype and gene expression data across multiple tissues. The estimated topics can be interpreted biologically and, in particular, they can be used to preselect genetic variant - gene pairs for eQTL testing.

The mathematical model underlying the method is carefully designed and the application on the GTEX v8 data looks convincing, as they detect many novel associations.

However, the manuscript itself could be improved, including a better description of the methods at multiple points and easier access for interested user without a strong statistical background. What remains unclear in the manuscript, is how simple an application would be for a potential user, regarding availability of code and estimated runtime.

Major Comments:

1. Making the code of the method available is crucial to enable other researchers to use their method.

We agree. We have added the public GitHub repository link to the Abstract.

2. The method is applied separately for each chromosome, both for genes and SNPs. Why did the authors choose this strategy? This will weaken the approach, as it cannot capture relationships across genes on different chromosomes and the same for SNPs on different chromosomes. If the reason is runtime or memory restriction, this should be stated in the manuscript. In general, reporting the runtime and memory requirements are very valuable information for potential users.

Thank you for this question. The correlation structure for SNPs is largely driven by linkage disequilibrium, which is not present across chromosomes. In terms of genes, we achieve the clearest separation of cis and trans associations by splitting genes into chromosomes. We have added the runtime and memory requirements into the manuscript.

3. The authors state that they don't want to model a gene expression-specific latent space in the fear of removing broad biological signal. But later they include PEER factors in their matrix eQTL analysis. Is that not contradicting with their initial reasoning? And if they want to include gene specific latent factor, it would be much more elegant to model them directly over their TBLDA model instead of using PEER afterwards (also easier for the user to not apply to many different methods for the eQTL analysis).

It is important to reiterate that the downstream eQTL analysis is meant for limited verification of 1:1 associations, and that the heart of the model is the latent topics themselves, which identify groups of associated features (a 'perpendicular' approach to univariate eQTLs). To determine whether traditional one gene-one SNP eQTLs are found within our topics (the aforementioned verification), we follow the GTEx consortium's trans-eQTL mapping pipeline, which includes PEER factors as covariates. In the manuscript, we additionally discuss the effects of running the univariate association tests without including PEER factors.

4. Linked to the last comment, the authors estimate the ancestry structure using their TBLDA model, but later for their matrix eQTL analysis, they do not include their own Ancestry Structure, but instead the top five genotype principal components. Why? Again, it would be nicer if the results from their model could be incorporated directly in the downstream eQTL analysis. In addition the number of eQTL discoveries for both approaches should be compared to each other (TBLDA genotype factors vs genotype PCs).

Thank you for the opportunity to clarify this. The ancestry structure portion of our TBLDA framework models entirely nonnegative factors. The downstream eQTL analysis runs linear regression-based eQTL mapping via Matrix eQTL. Genotype PCs fit the statistical assumptions of the regression model and it is important to use appropriate variables. Further, we follow the GTEx consortium eQTL mapping pipeline to ensure a similar methodological comparison, which includes genotype PCs as covariates.

5. The authors state that in the discussion that single-cell RNA-seq data would require a correction for batch effects, using a private subspace for gene expression data. But also bulk data can suffer from batch effects. Is there a way to apply their TBLDA model on bulk data with batch effects or will it only work on data without batch effects (which would be quite a limitation)?

For data with technical covariates, the user can simply identify the factors associated with those covariates (e.g. following our methodology to associate factors with tissue and cell type enrichment scores) and remove those factors from downstream analysis. In practice, this is the most straightforward approach to take batch effects into account using TBLDA. We have added this to Supp. File 1 to clarify options for future users.

6. The method for selecting the top "features" is not explained well (p.9, Feature Ranking): they first explain how they select the top 10% features and then the top 10% genes? Is the first part with 10% features including only SNPs? If yes, is there are way to determine the relative contribution of genetics and genes combined for one topic (so is the influence of genetics or gene expression stronger)? Please clarify notation and naming to avoid confusion.

Thank you for this comment. We have changed the first 'features' to 'SNPs' in the revised manuscript to clarify which modality we are referring to. The model does not have a way to determine the relative contribution of genetics vs gene expression.

7. Linked to the last comment, "tissue-associated SNPs" (p.9) are analyzed, but not tissue-associated genes, why?

We have added a description of tissue-associated genes to the revised Extended Methods (p. 9). Tissue-associated genes are discussed in the Results section.

8. Regarding the analysis of "robust components": for my understanding, robust components identify genes that are independent of association with SNPs and capturing often general tissue structure. Is this part of the model not hindering the identification of candidate eQTLs? Are the robust components somehow excluded when selecting candidate eQTLs?

While robust components often capture general tissue structure, they do not hinder the identification of candidate eQTLs. Factors with similar SNP weightings often cluster by tissue association across runs using different gene sets. This indicates that TBLDA can pull out common genetic signals active in particular contexts independent of particular genes. We use the sample data to interpret the latent factors by associating them with known covariates such as tissue. The robust components are not excluded when selecting candidate eQTLs. Although they are highly correlated, most robust components do differ slightly from run-to-run depending on the matched feature set used. This indicates that although common signal (such as tissue structure) is captured across runs, it is to an extent dependent on the features in the paired modality, which differ across chromosome-by-chromosome runs. Thus, we include these factors in the eQTL mapping.

9. The authors run first an eQTL analysis on the common factors. In contrast to the second step, the eQTL analysis on tissue-associated factors, they report here only the numbers without assessing their results compared to a full eQTL analysis. Do the eQTLs from the common factor approach also contain many novel eQTLs (as for the eQTLs from the tissue-associated factors)? The authors should show here if also the eQTLs analysis for common factors is a valuable selection strategy or if only the focus on tissue-associated factors brings new eQTL results?

We apologize for the confusion here– the main idea of modeling both data modalities is to discover latent factors that bridge them. For each learned factor, we test association between its top genes and SNPs (i.e. doing eQTL mapping for features loaded on a common factor). The tissue-associated factors are a subset of these factors that are associated with at least one labeled tissue type. We have added clarification in the eQTL results section.

10. The relation between cell type and LDA factors is very interesting. The authors should further explore if eQTL associated with these factors are enriched for cell type specific eQTL. This could be done exemplarily for whole blood, where both single cell (PBMC) and sorted bulk eQTL data sets (blueprint and immunexut) are available.

We agree that the possibility of cell-type eQTLs here is fascinating. We are currently extending these ideas to single cell data without restrictions on cell types and will leave those results to the follow-up manuscript. We have added this possibility to the Conclusion.

Minor Comments:

1. The model description is very mathematical, making it less understandable for readers without profound statistical background. A more abstract description of the model

combined with a better visualization than Figure 1 would improve the manuscript. Supplementary Figure 1 is already better showing the connection between the parameters (better than Main Figure 1).

Thank you for this comment. We have switched Fig. 1 with Supp. Fig. 1 to provide a better visualization to readers. Further, we reconstructed the revised Methods section to be much more readable, interspersing notation and variable definitions with the relevant equations. While we recognize that the model description is very mathematical overall, we feel that, as a methods paper, it is important that TBLDA be presented similarly to other articles in our field.

2. The authors report the known ancestry structure is not associated with shared factors (using correlation coefficient), but they do not report if the genotype-specific factors are correlated with the ancestry structure. Reporting for this some correlation value or similar would be valuable to prove that their genotype-specific subspace is indeed capturing the ancestry information.

We have added a sentence to the revised manuscript that addresses this.

3. Why was a cutoff of 40 chosen to define a tissue-specific factor?

A histogram of the inner products across all tissues and factors yields a heavy-tailed distribution (Supp. Fig. 2). While admittedly arbitrary, 40 appears to be a reasonable cutoff between factors with no/weak tissue associations versus strong tissue-factor associations.

4. Figure 5 is very difficult to read with all the tiny panels, the Figure 5 would benefit from a better visualization.

We have expanded Fig. 5 to span a full page so that it is easier to see.

5. The manuscript contains a section "Methods" and another section "Extended Methods". At multiple points, it says in the text "see Methods" while "see Extended Methods" would be correct.

Thank you for this comment. We agree and have changed all 'see Methods' to 'see Extended Methods'.

6. Was the LDSC annotation matched to the respective tissue? In other words were the SNPs indeed eQTL in the same tissue from which the enhancer annotations were obtained?

No, the LDSC file describes general annotations shared across tissues. We have added a sentence to clarify this in the revised manuscript. This is a caveat of our enrichment pipeline, but will likely create false negative enrichments rather than false positive enrichments.

Reviewer #3 (Comments to the Authors (Required)):

The authors are presenting an approach for eQTL mapping exploiting the fact that multiple expression measurements might exist per subject. Whereas multiple eQTL mapping approaches addressing this issue have been proposed, the authors here present a method that works on raw read counts assuming proper (discrete) distributions of the data.

I do see a need for such an eQTL mapping method.

However, I found the presentation of the method exceptionally hard to follow. I had asked two of my team members to join me in reading and discussing the manuscript. Therefore, I am going to use 'we' subsequently, because everything that follows is the result of intense discussions among the three of us.

We have a range of major concerns that we couldn't resolve despite our discussion and hence, we did not go beyond discussing the model itself and did not evaluate the results. We feel the manuscript has to be majorly re-written to make it much more accessible to scientists not working with LDA on a daily basis. All the following points refer to the presentation of the model, mostly equations 1-9, Figure 1 and in part also the Extended Methods at the end.

1. Information on the tissue the sample comes from does not seem to be used. We feel this is valuable information left out of the mapping. We fear that the algorithm may learn tissue context as topics (one topic per tissue), which would be an unnecessary learning step, 'wasting' degrees of freedom and potentially adding additional noise (e.g. if samples are assigned to the wrong 'tissue topic'). At least, this point should be addressed in the manuscript.

This is correct. We do not use tissue labels in our TBLDA model; instead, we use them post hoc to study possible tissue-specific associations. We have clarified this in the text. In our analysis of the "tissue context as topics", we refer the reviewer to Fig. 2 where we carefully show how the tissue specific topics are distributed across the different tissue types. In particular, samples from different tissues have a range of overlapping cell types, meaning that samples from certain tissues will share varying proportions of underlying processes and eQTLs. Additionally, in large data sets, some samples may be mislabeled. By allowing the model to learn common patterns and associations in an unsupervised manner and using the

tissue labels downstream, we uncover not only tissue-specific factors but also factors that associate with particular cell types shared across tissues.

2. The notation and indexing used in equations 1-9 is overly complicated. This is the biggest issue of the current manuscript. For example, samples are indexed with l and with i . Using l each sample is already uniquely indexed and hence the second index is redundant. Why this double indexing? This is just creating problems for understanding the equations.

We have removed the double indexing to avoid confusion.

3. Further, the notation between expression-related and SNP-related symbols is inconsistent. For example, the expression matrix is denoted with X and the genotype matrix is denoted with Y . Thus, different symbols are used to distinguish genotype and expression. However, the dimensionalities of the matrices are denoted with the same symbol (F) but different superscripts (F^g and F^s). This is creating a lot of downstream issues. For example, we were wasting a lot of time trying to understand equation (4). It was clear that equation (4) would compute a vector over genes. Yet, the superscript 'g' in x^g fooled us, because we thought this would be the expression level of a particular gene 'g' until we realized this was just indicating that x would be a vector of gene expression. Yet, in this case the superscript is not needed, because X was already introduced as the matrix of gene expression values. And so on

Thank you for pointing this out; in the revised text we have defined all variables introduced in Methods and replaced the symbols with superscripts to avoid such confusion.

4. The model plate diagram (Fig. 1) only partially helps, because it contains symbols that are not shown in equations 1-9. One has to refer to the Extended Methods to fully understand it.

We have clarified notation and swapped Fig. 1 and Supp. Fig. 1 to better illustrate the model.

5. We were discussing for a long time how and where correction for library size (read counts) is done, until we realized it is C_l , which is incorrectly denoted 'gene count' (should be 'read count'). Such mistakes further complicate the understanding.

We have corrected this typo, thank you. We note in the revised text that the total read count parameter is not fit as a 'correction' – it is set using the input sample data.

6. Equation 3 seems to average the expression level of a gene over the topics. Correct? Why can this be done? Averaging over multiple topic assignments for a single observation may

lead to infeasible averages. I.e. whereas either one or the other topic may be possible, what about the average?

This is an essential step in the telescoping model because there are multiple expression observations per single genotype. With TBLDA, the sample-topic proportions are largely spread over multiple topics and not fully weighted on any single topic. Given other related reviewer concerns, we think this point may be generally misunderstood in our manuscript. We completely revised the organization of our methods section to help clarify our framework.

7. Equation 7: is alpha user-defined or learned from the data? How is it set?

Alpha is learned from the data (Eqn. 7) and further discussed in Extended Methods and Supp. File 1. We have clarified this in the text.

8. Equation 8 needs better explanation. For example, what does the K with superscript s mean?

We have changed the notation from K superscript s to Q and defined Q above equation 8 to clarify this in the text.

9. "We include a modality-specific (private) subspace ..." Where is the product $\beta \cdot \zeta$ coming from? What does it mean?

The product $\beta \cdot \tau$ is learned from Terastructure and is described in Extended Methods—we have clarified this in the Methods section.

10. Extended Methods: For example in Ancestry Structure it says: "The resulting beta and theta output matrices were assigned ..." or in Model Runs: " ξ and σ_i were set to one, δ to 0.05, ...". In all of those cases it isn't clear what these parameters and output matrices are. One would have to read all the cited papers to understand their meaning, whereas the Methods should be self-explanatory.

Thank you, we have clarified this in Ancestry Structure. The hyperparameters described in Model Runs are from the equations from our own Methods section, which the user can refer back to. We have changed 'Model Runs' to 'TBLDA Model Runs' to make this more clear.

11. Stochastic variational inference: How would different parameters affect the results? Is there some motivation for the parameter choices? Are they simply the default values?

We only set six hyperparameters, and we have found our SVI approach robust to these values. The revised text includes Supplemental File 1, which provides insights to users about how to set the hyperparameters. We randomly initialize the variational parameters at the start of each TBLDA run.

July 8, 2022

RE: Life Science Alliance Manuscript #LSA-2021-01297-TR

Barbara E Engelhardt
Princeton University, Gladstone Institutes

Dear Dr. Engelhardt,

Thank you for submitting your revised manuscript entitled "Telescoping bimodal latent Dirichlet allocation to identify expression QTLs across tissues". We would be happy to publish your paper in Life Science Alliance pending final revisions necessary to meet our formatting guidelines.

- please address the remaining Reviewer #3 points
- the manuscript is in a different format, please take the columns out to allow for typesetting
- please incorporate the supp. Document into the Methods section in the main text
- please add the Twitter handle of your host institute/organization as well as your own or/and one of the authors in our system
- please consult our manuscript preparation guidelines <https://www.life-science-alliance.org/manuscript-prep> and make sure your manuscript sections are in the correct order; please add a separate section with your main and supplementary figures and your table legends
- please update your supplemental figures and label them with only the supplemental figure designation (e.g. Figure 7-Figure 11 should only be labeled as Fig. S1-S5)

A. FINAL FILES:

B. MANUSCRIPT ORGANIZATION AND FORMATTING:

Sincerely,

Reviewer #1 (Comments to the Authors (Required)):

The Authors have addressed our concrete concerns, but we feel it is also important to convince the other referees to make sure the intended audience is appropriately receptive to the contribution.

Reviewer #2 (Comments to the Authors (Required)):

All of my previous comments have been adequately addressed. Therefore I would recommend to accept the manuscript.

Reviewer #3 (Comments to the Authors (Required)):

The authors have considerably improved the presentation of their model. However, there are some remaining points.

Major points:

1. Still not all symbols from the equations are defined/named. Why are π_i and ρ_{ij} not explicitly defined, but only implicitly? This would greatly help the readability.
2. Tissue-specific factors: it is good news, that the model successfully identifies tissue-specific factors in an unsupervised approach. The identification of significantly enriched GO terms represents a truly independent validation. However, the same is not true for the TF analysis. The tissue-specificity of the TFs was determined by Sonawane et al. using the same data (GTEx). Hence, the consistency only demonstrates a consistent interpretation of the same data, but not a validation on independent data. That should be made explicit.
3. Why do trans-eQTL target an approximately equal number of non-coding and coding transcripts? To me this seems unexpected. May that be an indication of noise in the expression data of lncRNAs? There should be some discussion of this surprising finding.
4. Tissue and batch are not used as known covariates. I understand that including tissue (or batch or any other covariates) would further increase the complexity of the model. In their reply to my point 1 the authors discuss the advantages of their approach. I find that text very clear, why is it not included in the paper? I do think that there still are possible disadvantages, that should be discussed. I do not insist that the model itself be changed, but the pros and cons should be made more clear. Further, in their reply to Reviewer 2 the authors explain that batch can be corrected for in downstream analysis by identifying the respective factors (very much like for tissues). I think this statement should be included also in the main text (and not only in Supp. File 1).

Minor points:

5. Page 4: "... while only one gene set incorrectly included another tissue's TF ..." I would drop the 'incorrectly' in this sentence. It

is unclear which of the two studies is actually wrong (if any, maybe it is expressed in both tissues).

6. Page 5 "Notably, 48 regions on chromosome four were enriched for the robust SNP set associated with subcutaneous adipose" Isn't this chromosome 3? In addition, it seems that this region is enriched across multiple tissues. It would help to highlight that region e.g. with a dashed vertical line.

7. "This highlights the ability of TBLDA to identify jointly functional genomic regions even when the SNP data have been LD-pruned." What is the interpretation of these regions? Do they represent regions with multiple independently acting SNPs affecting the expression of co-regulated genes?

8. linc-RNA: I haven't heard of 'linc-RNA'. I think the authors mean long non-coding RNA, which is typically abbreviated as 'lncRNA'.

9. Figure 5: I think the upper part of this figure is overkill. The bottom two panels carry the same meaning, namely that more cis-eQTL have been found. Apart from that, the colors have different meaning in the upper and lower parts, which is confusing. If the upper panel is kept, the colors should be changed.

10. Figure caption 6: "Exploring the 34 trans-eGenes stemming from a single locus in thyroid." 'stemming from' sounds as if the genes were encoded at this locus, which I think is not the case. What about '... are targeted by a single locus ...'? Or 'associated with'?

11. eQTL Pipeline: why was the FDR corrected separately for coding and non-coding genes?

Major points:

1. Still not all symbols from the equations are defined/named. Why are π_l and ρ_{ij} not explicitly defined, but only implicitly? This would greatly help the readability.

We have now explicitly defined all symbols in the text.

2. Tissue-specific factors: it is good news, that the model successfully identifies tissue-specific factors in an unsupervised approach. The identification of significantly enriched GO terms represents a truly independent validation. However, the same is not true for the TF analysis. The tissue-specificity of the TFs was determined by Sonawane et al. using the same data (GTEx). Hence, the consistency only demonstrates a consistent interpretation of the same data, but not a validation on independent data. That should be made explicit.

This double-dipping into the GTEx data is a good point. We have changed the wording of the TF "validation" to the Reviewer's wording of "results consistent with the prior work" to make this explicit in the text.

3. Why do trans-eQTL target an approximately equal number of non-coding and coding transcripts? To me this seems unexpected. May that be an indication of noise in the expression data of lincRNAs? There should be some discussion of this surprising finding.

lincRNAs localize to the nucleus and are chromatin-associated, often acting in trans through chromatin modifiers (Cao et al. 2021; Hacisuleyman et al. 2014; Holdt et al. 2013). However, trans-eQTLs found using bulk RNA-seq can appear due to sample cell-type proportions (Vosa et al. 2021). Because lincRNAs show more cell-type specific expression than protein-coding transcripts, this could contribute to the imbalanced numbers of non-coding and coding trans eGenes (Grassi et al. 2021; Liu et al. 2016). We have revised the text to suggest this as a possible rationale for this surprising finding.

4. Tissue and batch are not used as known covariates. I understand that including tissue (or batch or any other covariates) would further increase the complexity of the model. In their reply to my point 1 the authors discuss the advantages of their approach. I find that text very clear, why is it not included in the paper? I do think that there still are possible disadvantages, that should be discussed. I do not insist that the model itself be changed, but the pros and cons should be made more clear. Further, in their reply to Reviewer 2 the authors explain that batch can be corrected for in downstream analysis by identifying the respective factors (very much like for tissues). I think this statement should be included also in the main text (and not only in Supp. File 1).

This is a great point, and we wanted to further clarify this in the manuscript. We have added the referenced text to the discussion, and placed the text from Supp. File 1 in the Extended Methods section.

Minor points:

5. Page 4: "... while only one gene set incorrectly included another tissue's TF ..." I would drop the 'incorrectly' in this sentence. It is unclear which of the two studies is actually wrong (if any, maybe it is expressed in both tissues).

We have dropped 'incorrectly'.

6. Page 5 "Notably, 48 regions on chromosome four were enriched for the robust SNP set associated with subcutaneous adipose" Isn't this chromosome 3? In addition, it seems that this region is enriched across multiple tissues. It would help to highlight that region e.g. with a dashed vertical line.

Thank you for pointing this out. This is chromosome 4, but the random shuffling of colors made chromosome 3 and 4 both the same shade of green. We have re-done this figure. We think the figure would look too busy with additional dashed lines added.

7. "This highlights the ability of TBLDA to identify jointly functional genomic regions even when the SNP data have been LD-pruned." What is the interpretation of these regions? Do they represent regions with multiple independently acting SNPs affecting the expression of co-regulated genes?

The genomic regions that are repeatedly identified contain multiple SNPs that contribute to variation of co-expressed genes within a particular tissue. TBLDA finds associations that are not necessarily directly involved in regulatory relationships, especially because the SNPs are LD-pruned, but they might be. We leave this exploration for downstream fine mapping and colocalization analyses.

8. linc-RNA: I haven't heard of 'linc-RNA'. I think the authors mean long non-coding RNA, which is typically abbreviated as 'lncRNA'.

We do mean lincRNA, which is commonly used to abbreviate long intergenic non-coding RNA. We have now spelled out this abbreviation explicitly in the text.

9. Figure 5: I think the upper part of this figure is overkill. The bottom two panels carry the same meaning, namely that more cis-eQTL have been found. Apart from that, the colors

have different meaning in the upper and lower parts, which is confusing. If the upper panel is kept, the colors should be changed.

We have changed the colors to differentiate the top and bottom figures.

10. Figure caption 6: "Exploring the 34 trans-eGenes stemming from a single locus in thyroid." 'stemming from' sounds as if the genes were encoded at this locus, which I think is not the case. What about '... are targeted by a single locus ...'? Or 'associated with'?

We have changed this to 'associated with'.

11. eQTL Pipeline: why was the FDR corrected separately for coding and non-coding genes?

This follows the GTEx trans-eQTL mapping protocol, which we were involved in developing. We separate out the coding and non-coding genes for the purposes of multiple-hypothesis testing correction.

July 18, 2022

RE: Life Science Alliance Manuscript #LSA-2021-01297-TRR

Prof. Barbara E Engelhardt
Gladstone Institutes
1650 Owens Street
San Francisco, CA 94158

Dear Dr. Engelhardt,

Thank you for submitting your Research Article entitled "Telescoping bimodal latent Dirichlet allocation to identify expression QTLs across tissues". It is a pleasure to let you know that your manuscript is now accepted for publication in Life Science Alliance. Congratulations on this interesting work.

DISTRIBUTION OF MATERIALS:

Again, congratulations on a very nice paper. I hope you found the review process to be constructive and are pleased with how the manuscript was handled editorially. We look forward to future exciting submissions from your lab.

Sincerely,
